# High Risk-Human Papillomavirus in HNSCC: Present and Future Challenges for Epigenetic Therapies

**DOI:** 10.3390/ijms23073483

**Published:** 2022-03-23

**Authors:** Lavinia Ghiani, Susanna Chiocca

**Affiliations:** Department of Experimental Oncology, IEO, European Institute of Oncology IRCCS, IEO Campus, Via Adamello 16, 20139 Milan, Italy; lavinia.ghiani@ieo.it

**Keywords:** Head and Neck Squamous Cell Carcinoma (HNSCC), head and neck cancer (HNC), Human Papillomavirus (HPV), epigenetics, histone post-translational modifications (hPTMs), therapies, sex, gender

## Abstract

Head and Neck Squamous Cell Carcinoma (HNSCC) is a highly heterogeneous group of tumors characterized by an incidence of 650,000 new cases and 350,000 deaths per year worldwide and a male to female ratio of 3:1. The main risk factors are alcohol and tobacco consumption and Human Papillomavirus (HPV) infections. HNSCC cases are divided into two subgroups, the HPV-negative (HPV−) and the HPV-positive (HPV+) which have different clinicopathological and molecular profiles. However, patients are still treated with the same therapeutic regimens. It is thus of utmost importance to characterize the molecular mechanisms underlying these differences to find new biomarkers and novel therapeutic targets towards personalized therapies. Epigenetic alterations are a hallmark of cancer and can be exploited as both promising biomarkers and potential new targets. E6 and E7 HPV oncoviral proteins besides targeting p53 and pRb, impair the expression and the activity of several epigenetic regulators. While alterations in DNA methylation patterns have been well described in HPV+ and HPV− HNSCC, accurate histone post-translational modifications (hPTMs) characterization is still missing. Herein, we aim to provide an updated overview on the impact of HPV on the hPTMs landscape in HNSCC. Moreover, we will also discuss the sex and gender bias in HNSCC and how the epigenetic machinery could be involved in this process, and the importance of taking into account sex and/or gender also in this field.

## 1. Head and Neck Squamous Cell Carcinoma

Head and Neck Squamous Cell Carcinoma (HNSCC) is a highly heterogeneous group of tumors arising in the epithelial cells of mucosal linings of different anatomical sites of the head and neck district, including paranasal sinuses, nasopharynx, lips, oral cavity, oropharynx, larynx and hypopharynx, characterized by different cell type composition and tissue organization [1].

Approximately 90% of all head and neck cancers belong to HNSCC. Indeed, it is characterized by an incidence of more than 650,000 new cases and 350,000 deaths per year worldwide. It is more frequent in males than females, with an incidence ratio approximately equal to 3:1. Moreover, it is generally diagnosed at an average age of 50–70 years [1,2,3,4].

Despite the therapeutic treatments consisting mainly of surgery, chemotherapy, radiotherapy and, more recently, immunotherapy, the 5-year overall survival is approximately 66%. Globally, the major clinical hurdles include the presence of distant metastases in 10–30% of HNSCC cases and tumor relapse in 30–50% of patients, often leading to therapy resistance [5,6].

### 1.1. HNSCC Risk Factors and HPV Detection

Histologically, HNSCC onset occurs in different steps: it starts with the development of epithelial cell hyperplasia, followed by dysplasia, carcinoma in situ and invasive carcinoma [4,6].

Alcohol consumption, smoking and poor oral hygiene are the main risk factors involved in pathology development. In the last decades, high-risk Human Papillomavirus (hr-HPV) has also emerged as another important etiological factor [1,6]. Thus, HNSCC are subdivided into two main subgroups: the HPV negative (HPV−) and the HPV positive (HPV+). Importantly, since HPV− and HPV+ tumors display a large plethora of molecular and clinicopathological differences, they are recognized as distinct entities (Table 1) [1].

HPV+ HNSCC represent approximately 25% of the worldwide HNSCC cases, and their incidence is different depending on the tumor anatomical site. The highest levels are observed among Oropharyngeal Squamous Cell Carcinomas (OPSCC): nearly 70% of HPV+ HNSCC cases occur in the oropharynx and approximately 60% to 70% of OPSCC are HPV+ [1,6,7]. One of the main reasons for this distribution is probably due to the discontinuous single-layered epithelium organization that characterizes the oropharyngeal region and that is more prone to carcinogenic transformation compared to the epithelia of other anatomical sites [1,4]. HPV-based HNSCC classification has been recently officially accepted with the publication of the 8th edition of the AJCC (American Joint Committee on Cancer), where the stage classification takes into consideration the HPV status for OPSCC [8]. This underlies the importance to precisely assess whether the tumor is HPV-driven through specific HPV diagnostic assays [9,10]. HPV positivity is assessed via multiple approaches, leading to either over- or under-estimate the number of tumors that are effectively HPV+. The gold standard for HPV detection is through Immunohistochemistry (IHC) of the p16INK4a surrogate marker, where a percentage of p16-positive staining ≥70% is associated to HPV positivity. However, recent studies have shown that this biomarker use could be misleading, since approximately 20–30% of HPV− HNSCC cases have been found positive to p16 [9]. Thus, strategies based on viral DNA detection and on E6*I mRNA levels, a marker of viral oncogene transcription activity, should be preferred [4,6,9]. The combinations of multiple biomarkers for assessing HPV-driven cancer have also been reported [10,11].

Although the incidence of HPV− HNSCC has been decreasing over the last fifty years in the USA and Western Europe due to reduced smoking consumption, new HPV+ OPSCC cases have significantly increased and projections suggest further increase in the next years [12,13]. Epidemiological data suggest that the Gardasil 9 prophylactic vaccine against HPV, approved by the FDA (Food and Drug Administration) in 2014 in substitution to the previously approved Gardasil vaccine (approved in 2006 for genital cancers), is expected to reduce the HPV+ OPSCC incidence not earlier than 2060. This delay is due to the 10–30 years latency period between HPV infection and clinical manifestations of HPV+ OPSCC (https://www.science.org/, accessed on 21 March 2022, https://www.esmo.org/, accessed on 21 March 2022) [1,13]. Importantly, in 2018 vaccine administration was extended, upon FDA approval, to both men and women between 27 and 45 years (https://www.fda.gov/news, accessed on 21 March 2022). In June 2020 Gardasil9 has been also approved by FDA for the prevention of HPV related HNSCC with an efficacy that has been estimated ranging from 88–93% (https://www.merck.com/news/, accessed on 21 March 2022) [14].

### 1.2. HPV+ and HPV− HNSCC Are Distinct Tumoral Entities

HPV+ HNSCC usually occur in younger patients (6th decade of life), have higher immune infiltrate, are less differentiated, present better responsiveness to conventional therapies and display an overall more favorable prognosis compared to the HPV− ones (5-years survival rate of 75–80% versus ~55% in HPV−) (Table 1) [1,13]. In this context, several aspects are however still debated: for example, the effect of the interaction between viral infection and other risk factors (e.g., alcohol and tobacco) as well as the prognostic value of HPV infection in non-oropharyngeal HNSCC (laryngeal, oral or hypopharynx). Indeed, HPV+ laryngeal cancers show worse prognosis compared to the HPV+ oropharyngeal cancers [11].

Moreover, molecularly, HPV+ and HPV− HNSCC are significantly different. Comprehensive genomic and transcriptomic analyses revealed deep differences occurring in their mutational and transcriptional profiles (Table 1) [1,15,16,17,18]. The higher mutational burden in HPV− compared to HPV+ HNSCC could be explained by carcinogens derived from tobacco and alcohol consumption. They induce DNA adducts formation, leading to hypermutations and chromosomal instability, and, finally, to tumorigenesis. On the other hand, within HPV+ HNSCC, genetic and transcriptional alterations are predominantly induced by viral infection through two main approaches: (i) a crucial role is exerted by the oncoviral proteins E6 and E7 that dramatically impair the host cell homeostasis by interacting and destabilizing a large number of host proteins, and (ii) linearization and integration of viral DNA in the host genome lead to genomic instability with genomic rearrangements such as amplifications, deletions, inversions and chromosomal translocations [1,15]. Beyond the differences in their genomic and transcriptomic profiles, HPV+ and HPV− HNSCC are also characterized by different epigenetic landscapes. Distinct DNA methylation profiles within the two subtypes are very well documented, with HPV− HNSCC being characterized by a global hypomethylated status compared to the HPV+ cases. Moreover, different enrichments in histone post-translational modifications (hPTMs) between the two subgroups are also reported [19,20,21,22,23].

## 2. HPV and Its Impact on the Host Cells in HNSCC

Despite the well-known HPV causal role in the onset of cervical and anogenital cancers [24], the link between HPV and HNSCC was only firstly documented at the end of nineties [25].

HPV belongs to the *Papillomaviridae family* with over 200 subtypes divided in low- and high-risk (hr) HPV. Thirteen high-risk types have been characterized: HPV16, HPV-18, HPV-31, HPV33, HPV35, HPV39, HPV45, HPV51, HPV52, HPV56, HPV58, HPV59 and HPV68 [26].

HPV16 and HPV18 are the main high-risk types responsible for both cervical and HNSCC, with HPV16 accounting for 90% of HPV+ OPSCC [27].

The HPV genome is organized as a double-stranded DNA spanning from 6.8 to 8 kb in length. It is composed by three main regions: the early gene-coding region (E), encoding for E1, E2 and E4–E7; the late gene-coding region (L), encoding for the viral capsid proteins L1 and L2; the long control region (LCR), controlling DNA viral replication and transcription. E5, E6 and E7 are the main oncogenic proteins and the difference between low- and high-risk HPV types lies in the different affinity for their targets [26].

HPV infects basal cells of stratified squamous epithelium, both of cutaneous and mucosal tissues, mainly of hands and feet, anogenital and upper aerodigestive tracts, generally taking advantage of micro-abrasions or wounds. Most infections are permissive for viral replication and spontaneously resolved, mainly thanks to specific cell contexts and immune responsiveness, while others give rise to malignant transformation. For example, only 3–5% of infected cervices lead to cellular transformation [1,4].

E1 and E2 promote viral replication at low copy, taking advantage of the host cell replicative machinery. The virions are released upon host cells differentiation. In detail, basal infected cells differentiate into the suprabasal layer, and, as soon as epithelium desquamates, virions are spread and are ready to infect other cells. Normally, the host-immune response arrest viral propagation. However, the defense mechanism may be evaded in transforming infections, due to effects of oncoviral proteins E6 and E7 [1,26].

E6 and E7 transcription is normally repressed by E2. Upon viral infection, HPV genome can integrate itself in the genome of the host cell or, alternatively, can persist in an episomal status. HPV genome integration is a crucial event for malignant transformation and it is responsible for the viral cycle blockade. During the integration process, HPV genome breakpoint occurs mainly within the *E2* gene, causing its disruption and leading to E6 and E7 oncoproteins overexpression [26]. When the HPV genome is instead found in the episomal status, E2 expression is repressed by DNA methylation at its promoter region [26]. The expression of the viral oncogenes E6 and E7 are exploited by HPV to deregulate the host cell replicative machinery, in order to replicate its genome.

E6, by recruiting the E6AP ubiquitin ligase, leads p53 to proteasomal degradation, thereby abrogating the p53-mediated apoptotic pathways; on the other hand, E7 induces the retinoblastoma protein (pRb) proteasomal degradation, leading, in turn, to E2F genes’ activation and S-phase entrance. Apart from p53 and pRb, E6 and E7 also interact with several other host proteins including ubiquitin ligases, transcriptional factors and epigenetic regulators, thus promoting the oncogenic reprogramming of the host cell (Figure 1).

Overall, E6 and E7 promote uncontrolled cell division, replication of both the host and viral genomes, immortalization and genome instability [1]. They also induce overexpression and misfunction of genes involved in DNA damage repair, a mechanism exploited by HPV to expand itself within the host genome and to notably increase genome instability [26]. Furthermore, E6 and E7 affect tumor antigen presentation, avoiding immune cell recognition and, consequently, promoting tumor immune evasion [26].

Thus, as an example, within HPV+ tumors, genomic rearrangements such as amplification of oncogenes (e.g., *hTERT*, *TP63* isoform *ΔNp63α*) or deletions in regions encoding for genes (e.g., *TRAF3*) have been well described [26].

E5 is the other viral oncoprotein encoded by the HPV genome. It is a transmembrane protein that prevents apoptosis and promotes cellular proliferation, immortalization and cellular transformation. However, its role seems only implicated in initial stages of carcinogenesis, and not in persistence. This observation is supported by the E5 lack in host genome upon viral integration [26].

Overall, this evidence demonstrates E6 and E7 crucial role during oncogenic process, and the high complexity of the HPV-related tumoral phenotype.

### The HPV Impact on the Epigenetic Pathways of the Host Cells

E6 and E7 regulate the expression of several micro-RNAs and the expression and activity of a large number of epigenetic factors. As mentioned, oncoviral proteins impair host cell epigenetic landscape, significantly altering both DNA methylation and hPTMs profiles [19,28]. This remodeling has a crucial impact on genes’ expression alterations and is implicated in malignant transformation. Investigating the epigenetic alterations occurring in HPV+ HNSCC and understanding their role in cancer progression and in therapeutic resistance and recurrences is needed for the discovery of new clinical biomarkers, of novel effective and promising epigenetic targets, and for their positioning in personalized medicine.

## 3. Epigenetics in Cancer

Alterations of epigenetic profiles are crucial events for cancer onset and progression. Epigenetic modifications include DNA methylation, hPTMs, nucleosome positioning, regulation of non-coding RNA and post-transcriptional mRNA modifications [29]. Most of the epigenetic modifications regulate transcription, through chromatin accessibility modulation. Epigenetic modifications are reversible events. Many enzymes, called epigenetic modifiers, alter chromatin status: the so-called “writers” deposit chemical groups on specific residues of histone tails; the “erasers” remove the deposited groups; the “readers” specifically recognize and bind epigenetic marks, playing an effector role in different processes. Thus, epigenetic modifications, by recruiting other proteins with different functions, are implicated in the regulation of several biological processes such as mRNA splicing, transcriptional elongation, DNA damage repair, DNA recombination, replication and X-chromosome inactivation [29,30].

Importantly, mutational events, copy number alteration or aberrant expression levels of epigenetic modifiers have been reported in several cancers, and are involved in the activation of many oncogenic pathways such as cell proliferation, apoptosis, cell migration, tissues invasion, metabolic reprogramming, differentiation, clonogenicity and immune evasion. Moreover, epigenetic remodeling influences also acquired drug resistance, implying a clear effect in acquired-drug resistance [29].

Gain-of-function mutations or altered expression of genes encoding metabolic enzymes represent another mechanism through which epigenetic profiles are affected by deregulating metabolic pathways. Several metabolites indeed represent cofactors, substrates or competitive inhibitors of epigenetic enzymes. For example, mutations in isocitrate dehydrogenase (IDH) and overexpression of nicotinamide N-methyltransferase (NNMT) enzymes occur in cancers, including HNSCC, promoting cancer progression and chemoresistance, and induce alterations in DNA and histone methylation profiles [31,32,33,34,35,36].

Compared to normal cells, cancer cells are more addicted to some specific epigenetic regulators. While in normal contexts their inhibition is compensated by semi-redundant mechanisms, in cancer cells they are required to control the expression of few sets of genes and are essential. This is a phenomenon known as “epigenetic vulnerability” and is, in turn, based on the concept of “oncogene addition”, consisting in tumor cell dependency on specific oncogenes or pathways to maintain the malignant phenotype [29,37]. Thus, targeting epigenetic regulators is emerging as a promising effective anticancer therapeutic option, alone or in combination with other drugs. Pre-clinical studies and clinical trials have shown that the use of epigenetic drugs (epi-drugs) in combination with chemo- and radiotherapy can significantly improve patients’ outcome [38]. Moreover, epi-drugs can boost anti-tumor immune response promoting T-cell attraction, enhancing immune checkpoint inhibitor efficacy and, more in general, immunotherapy response, and this makes them even more attractive [30,38,39]. A large amount of emerging evidences seems to suggest that this could be applicable also for HNSCC, a disease where epigenetic therapy is still at its initial discovery stages [40].

Epi-drugs are divided into two main groups: the so called “broad reprogrammers” and the “epigenetic target therapies”. The “broad reprogrammers” include Histone Deacetylases (HDACs) and DNA Methyltransferases (DNMTases) inhibitors and induce broad changes in gene expression reverting cancer phenotype. The “epigenetic target therapies” are potentially designed for cohorts of patients carrying specific activating mutations affecting the epigenetic pathways [38]. To date, only “broad reprogrammers” epi-drugs are in clinical trials for HNSCC treatments [20,30].

### 3.1. DNA Methylation

DNA methylation is one of the most characterized epigenetic pathways impaired in HNSCC. DNA methylation occurs on cytosine residues and generate 5-Methylcytosine (5 mc) in CpG dinucleotides, a repressive mark predominantly found in correspondence of heterochromatin regions such as centromeres, telomers, inactivated X-chromosome and at the level of repeated sequences and inactive promoters [29]. CpG islands are long stretches of CpG dinucleotide-rich regions located in approximately 60% of mammalian promoters and have a repressive transcriptional role [41]. Interestingly, 5–10% of physiologically unmethylated CpG islands lying on promoters are aberrantly methylated in cancers, leading to transcriptional repression of tumor suppressors [29,42]. DNMT1, DNMT3a and DNMT3b are the most studied DNA methyltransferases: while DNMT1 is mainly involved in methylation status maintenance, DNMT3a-b are involved in the de-novo deposition of methyl-groups on DNA [29,42].

DNA methylation pattern differences between HNSCC and normal tissues, as well as between HPV+ and HPV− HNSCC, have been widely described.

E6 and E7 induce DNMT1, DNMT3a and DNMT3b overexpression resulting in global DNA hypermethylation [43,44]. HPV+ HNSCC has been associated with DNA hypermethylation compared to normal tissues and to HPV− HNSCC, which on contrary, are characterized by global DNA hypomethylation. Studies focusing on OPSCC show that DNA hypermethylation occurs both within coding regions and Transposable Elements (TEs) such as LINE-1, Alu and LTRs. Moreover, regarding HPV+ HNSCC, a specific 5-CpG methylation signature has been identified [21,22,44].

Importantly, this characterization is crucial for the discovery of novel therapeutics targets and represents a significant step towards the identification of new biomarkers for early diagnosis. Indeed, the lack of screening systems is one pitfall in HNSCC management during late-stage diagnosis. Thus, analysis of the methylation profiles of DNA extracted from saliva or oral rinses could represent new promising biomarkers for early detection, prognosis and diagnosis of HNSCC [45].

### 3.2. Histone Post-Translational Modifications (hPTMs)

Differently from DNA methylation patterns, hPTMs have not been mapped out in detail in HPV+ and HPV− HNSCC.

hPTMs are chemical residues that are deposited and/or removed by specific epigenetic modifiers at histone N-terminal tails protruding by the nucleosome, an octamer composed by two copies of each core histone (H2A, H2B, H3 and H4), around which are wrapped 147 bp of DNA. Chemical modifications on histones residues alter non-covalent interactions within and between nucleosome and DNA, resulting in structural chromatin changes [29,46]. Several hPTMs, such as phosphorylation, ubiquitination, sumoylation, ribosylation, crotonylation and serotonylation, have been described in literature [46].

However, the most studied and characterized are histone acetylation and histone methylation: in this review we will focus on these hPTMs [46].

#### 3.2.1. Histone Acetylation

Histone acetylation occurs mainly on H3 and H4 histones. By neutralizing the positive charge of lysines and reducing the interactions between nucleosomes and DNA, histone acetylation induces an open chromatin conformation leading to transcriptional activation [47]. Accordingly, it has been widely described that acetylated residues lie on active promoters, active enhancers and gene body regions. Histone acetylation results from the activity of histone acetyltransferases (HATs) and HDACs. Altered expression and/or activity of HATs and HDACs have been described in a wide range of malignancies [29]. Despite the low number of mutations reported on HDAC genes, they are often aberrantly expressed in malignancies. Hyperactivation or overexpression of HDACs causes a global reduction of histone acetylation which reprograms cellular homeostasis and represses the expression of given genes, such as tumor suppressor genes [29,47].

#### 3.2.2. Histone Methylation

Histone methylation is catalyzed by histone methyltransferases (HMTs) which deposit mono-, di- or tri-methyl groups on arginine, lysine and histidine residues of H3 and H4 histone tails; methyl groups removal is instead orchestrated by histone demethylases (HDMs). Contrarily to histone acetylation, histone methylation does not affect the overall charge of the nucleosome and is associated with transcriptional activation and euchromatin conformation as well as to transcriptional repression and heterochromatin conformation. In humans, several “canonical” lysine methylation sites have been described, such as H3K4, H3K9, H3K27, H3K36, H3K79 and H4K20; in addition to these, several others less characterized are referred as “non-canonical” such as H3K23me, H4K5me1 and H4K12me [29]. Many of the canonical marks have been classically associated to active transcription (H3K4, H3K36 and H3K79) while others to transcriptional repression (H3K9, H3K27 and H4K20) [29,47]. According to the genomic loci, to the crosstalk with other histone marks and to the balanced activity with epigenetic cofactors, each of these histone marks can be associated either to repressive or activating role [47].

Lysine methyltransferases (KMTs) can have a redundant role, meaning that an enzyme can methylate more than one substrate (e.g., SETDB1 has two described targets, H3K4 and H3K9), but they can also be very selective for their substrates methylating only a histone residue and with a specific grade of methylation. For example, H3K36me1/me2 are specifically produced by proteins of the NSD family and by ASH1L, while SETD2 is the unique known enzyme responsible for H3K36me3 [48]. HMTs and HDMs are often mutated or aberrantly expressed in cancers [29].

## 4. The HPV Impact on the hPTMs Landscape of Host Cells in HNSCC

Dissecting hPTMs profiles in cancer is an emerging strategy with important implications in different directions such as discovery of new specific biomarkers for patients’ stratification, and for identification of new therapeutic targets. In this context, the technological advances in novel and more accurate quantitative mass-spectrometry based approaches represent a powerful tool for a fine profiling of the epigenetic landscapes in cancers [49,50,51].

As mentioned, clear and unambiguous characterization of the hPTMs profiles distinguishing HPV+ from HPV− HNSCC is still missing. For hPTMs the emerging scenario is quite complex and intricate, and, differently from DNA methylation, few studies profiling the differences have been published. This can be explained by the lack of unambiguous way in which HNSCC studies are designed. In some works, HNSCC cases are classified all together, without any stratification based on HPV status or anatomical region, and compared to normal tissues. On the other side, other studies divide HNSCC cases in subtypes, each with different criteria: some according to the anatomical subsite, others according to the HPV status, others again subdividing the HPV+ cases by anatomical site [19,22,52]. Another issue is related to the different technical approaches used in these studies and to their low or different intrinsic sensitivity and/or specificity. Data are commonly generated via molecular assays (such as IHC and Western blot) performed on HNSCC derived cell lines, tissue samples, or normal primary keratinocytes transduced with the E6 and E7 oncoviral proteins. To our knowledge, only one study reports the characterization of hPTMs in HNSCC patients through the use of super-SILAC mass-spectrometry [53]. However, it includes only a comparison between normal and tumoral HNSCC, without any distinction on the HPV status. Specifically, global higher levels of H3K36me2, H3K36me1 and of the dipeptide H3K27me1/K36me1, H3K27me1/K36me2 and lower levels of H3K14ac, H3K27me3 and of the dipeptides H3K27me3/K36me1, H3K27me2/K36me1 were found in HNSCC compared to normal tissues [53].

Overall, some published results on HPV- and HPV+ HNSCC hPTMs are summarized in Table 2, and in some cases they look contradictory. In 2017 Biron et al., demonstrated by IHC that HPV+ (p16-positive) OPSCC have higher levels of H4K20me1 and of H3K27me3 and lower levels of H4K20me3 in comparison to HPV− (p16-negative) OPSCC [54]. However, regarding H3K27me3 other studies show the opposite trend. In particular, they suggest that oncoviral proteins E6/E7 lead to H3K27me3 strong reduction [55]. Interestingly, E7 induces overexpression of both EZH2 and KDM6A, suggesting that the overall readout is the result of a balance among the effect of multiple histone modifiers [55,56,57,58]. Moreover, it is always very important to consider the H3K27me3 status associated to specific genes: studies have demonstrated that, besides H3K27me3 global levels, a selective loss of this histone mark occurs at the level of specific promoters, leading to the silencing of their controlled genes [59].

In an initial study describing HNSCC chromatin organization through ChIP-seq, the two active marks H3K27ac and H3K4me3 were correlated with the expression of tumor-specific genes and described as highly disease-specific histone marks having an elevated tissue-type specific genome-wide distribution [61].

The majority of the studies aimed at characterizing the HPV impact on the epigenetic pathways of the host cell are focused on the effect of the E6 and E7 oncoviral proteins on the expression levels and activity of a large number of epigenetic regulators. It is important to note that also HPV DNA integration in the host cell genome is emerging to alter hPTMs distribution. This is the case for H3K27ac: genomic regions in correspondence of HPV integration sites occurring often on enhancers regulating genes implicated in HNSCC tumorigenesis (such as *TP63*, *NOTCH* and *FOXE1*), are enriched for H3K27ac and thus activated [61]. Moreover, it has been recently demonstrated that regions enriched for H3K27ac are also significantly associated with cancer-specific alternative splicing events (ASEs). These occur prevalently in HPV+ OPSCC and contribute to the oncogenic reprogramming of the host cell [62]. Interestingly, treatment of HPV+ OPSCC cell lines with JQ1, an inhibitor of BRD4 (an epigenetic reader of acetylated histone residues), downregulates ASEs expression and inhibits cell growth, highlighting the role of hPTMs alterations in the oncogenesis of HPV+ OPSCC and indicating a novel promising epigenetic target for these tumors [62].

Analysis of RNA-seq data from publicly available TCGA datasets, published in [53], shows how the expression levels of several histone modifiers are altered in HNSCC samples. This is the case for *HDAC2, KDM5A, KDM4A, KDM3A, KDM5B, KDM1A* and *EHMT1*, that are upregulated, and for *SIRT2, HDAC6, HDAC5, KAT2B, SETD2, SETD3* and *SMYD1*, that are downregulated in HNSCC compared to normal samples [53].

Besides all these observations, several studies report the impact of HPV mainly mediated by the E6 and E7 oncoviral proteins on the epigenetic machinery of the host cell. Here we discuss some of the known main mechanisms, including histone acetylation and methylation.

### 4.1. HPV and Histone Acetylation

HPV E6 and E7 oncoviral proteins interact with both HATs and HDACs both modulating their activity and expression levels (Table 3).

#### 4.1.1. HPV and Acetyltransferases

- **p300 and its paralogue CBP (p300/CBP)** are HATs and transcriptional coactivators that acetylate all four histone cores as well as several other non-histone targets. They are modulated by E6 and E7 oncoviral proteins: E6 interacts with p300/CBP reducing its activity [62,63]. Moreover, this interaction inhibits the p300/CBP-mediated p53 acetylation and consequently the activation of p53-dependent gene expression, contributing to E6 induced cellular transformation [63,64]. E7 promotes the acetylation of pRb forming a multimeric protein complex with both p300/CBP and pRb. pRb acetylation reduces pRb phosphorylation and thus its inhibitory role in cell cycle progression [65,76]. Interaction between E7 and p300 also inhibits the interaction between p300 and the HPV E2, that increases E2-mediated transcriptional expression of viral genes [76].

**- PCAF (p300/CBP-Associated Factor)/KAT2B** is a HAT that associates with p300/CBP. It acetylates the H3 and H4 histones and preferentially H3. E7 interacts with PCAF impairing its acetyltransferase activity and this mechanism is thought to be involved in cell cycle deregulation and de-differentiation occurring in HPV-infected cells [66].

**- TIP60** is a HAT and a tumor suppressor. It acetylates histones H4 and H2A and non-histone targets, many of which involved in DNA double strand break (DSB) response pathways [67]. Both low- and high-risk HPV E6 interact with TIP60 and act as adaptors to facilitate the interaction with cellular ubiquitin ligases, thus promoting TIP60 proteasomal degradation. TIP60 degradation seems to be involved in the impairment of checkpoint activation in p53-mediated apoptotic pathways and in the deregulation of the differentiation program. Interestingly, TIP60 acetylates histone H4 that is then recognized by BRD4. The latter represses HPV E6 transcriptional activation. Thus, it seems that for both low- and high-risk HPV targeting TIP60 is an important step to sustain viral life cycle and cellular deregulation observed both in malignant and benign HPV induced papillomas [68].

**- GCN5/KAT2A** is the first identified Lysine Acetyltransferase (KAT) and is a catalytic component of the SAGA complex and acetylated histones H3, H4 and H2B as well as non-histone targets [69,70]. It has been recently demonstrated that E7 induces the upregulation of GCN5. This upregulation induces cell cycle progression both by promoting histone acetylation at the E2F1 promoter and by acetylating c-Myc that thus gains higher affinity for the E2F1 promoter itself [77].

#### 4.1.2. HPV and Histone Deacetylases

In humans there are 18 Histone Deacetylases (HDAC) enzymes divided into four classes: Class I (HDAC1, HDAC2, HDAC3, HDAC8); Class II (HDAC4, HDAC5, HDAC6, HDAC7 and HDAC9); Class III (SIRT1, SIRT2, SIRT3, SIRT4, SIRT5, SIRT6 and SIRT7); the Class IV protein (HDAC11) [29].

**- HDACs** have been widely described to interact with or to be regulated by HPV E6 and E7 oncoviral proteins. The interaction between E7 and HDACs induces the activation of cellular promoters, some of which are involved in cellular differentiation and are necessary for specific phases of the virus life cycle [28]. E7 indirectly interacts with HDAC1 and HDAC2, significantly affecting the transcriptional program of the host cell. HDACs are necessary for Hypoxia inducible factor 1α (HIF-1α) activity, which is high in HPV+ tumors. Indeed E7, by displacing HDAC1, HDAC4 and HDAC7 from HIF-1α, enhances HIF-1α dependent transcription, thus regulating hypoxic responses [71]. E7 also recruits HDAC1/2 to the tumor suppressor IRF-1, thus inhibiting IRF-1 transcriptional activation and the production of IFN-β, suggesting a possible mechanism underlying immune evasion, frequently observed in HPV+ cancers [72]. Moreover, E7 has been demonstrated to increase H3K9 acetylation levels on the promoters of E2F-responsive genes, leading to transcriptional activation [78].

HPV has been also demonstrated to indirectly regulate the expression levels of HDAC6: E6 downregulates miR-22 that targets HDAC6, a cytoplasmatic deacetylase involved in oncogenic pathways [79].

**- Sirt1** is a member of the HDAC Class III, and it deacetylases the histones H1, H2A, H3 and H4, as well as the non-histone targets p52 and FOXO proteins [73]. It has well described oncogenic functions, and is upregulated by the HPV E7 oncoprotein [74], thus representing one of the mechanisms of E7 driven oncogenic transformation. Interestingly, Sirt1 is involved in the regulation of HPV16 E1-E2-mediated DNA replication [75].

### 4.2. HPV and Histone Methylation

HPV E6 and E7 oncoviral proteins also interact with, affect the activity or the expression levels of several histone methyltransferases or demethylases (Table 4).

#### 4.2.1. HPV and Methyltransferases

**- EZH2** is the catalytic subunit of the Polycomb Repressive Complex 2 (PRC2) that catalyzes H3K27me3 methylation. Its overexpression in HNSCC is well documented and associated with poor prognosis [80,81,82]. HPV+ HNSCC have higher levels of EZH2 compared to the HPV− [83]. In HPV+ HNSCC, the overexpression of EZH2 is induced by both E6- and E7-mediated mechanisms and required for cell proliferation and other oncogenic functions [58]. As discussed in Section 5, EZH2 is considered as a promising target for the treatment of HNSCC although further investigations are required to evaluate whether HPV+ and HPV− HNSCC patients could respond differently to EZH2 inhibitors treatments [82]. Some evidence indeed suggests that HPV+ cases could respond better than the HPV− ones [83].

**- SUV39H1** is a H3K9me3 methyltransferase associated with chromatin closed conformation. Its expression levels are increased by hr-HPV E7. Higher levels of SUV39H1 suppress cGAS-STING expression, which is part of the signaling axis involved in the recognition of cytoplasmic DNA and in immune response activation [84]. SUV39H1 expression levels seem also to induce higher levels of DNMT3A, thus participating in E7-mediated upregulation of DNA methylation and in the intricate crosstalk existing between hPTMs and DNA methylation [93].

**- NSD protein family members (NSD1, NSD2 and NSD3)** are H3K36 methyltransferases that catalyze the mono- and di-methylation of histone H3 lysine 36 (H3K36me1/H3K36me2) in a non-redundant manner [94]. They have been widely associated with oncogenic properties in several cancers among which HNSCC [94]. Few data have been published on their function in HPV+ HNSCC: evidence suggests that NSD2 is upregulated in HNSCC independently from the HPV status [60,94]. However, a recent report highlights that NSD1, NSD2 and NSD3 proteins are upregulated in the HPV+ compared to the HPV− tumors. The authors show that low expression levels of these enzymes are associated with poor survival in HPV+ HNSCC while not in the HPV− ones [85]. Thus, further studies are needed to understand the role of these proteins in HNSCC.

**- SET7** is a methyltransferase that methylates H3K4, H3K37 and H4K20 histone marks [95,96]. It has been demonstrated to interact with HPV E6, inhibiting its methyltransferase activity [86]. As other methyltransferases, SET7 has also non-histone targets, such as p53. Upon methylation, SET7 stabilizes p53, thereby preventing its E6 mediated degradation [86]. However, at the epigenetic level, the effects of SET7 modulation in HPV+ HNSCC have not been investigated in depth.

#### 4.2.2. HPV and Histone Demethylases

**- KDM2A** demethylates the mono- and di- methyl group from H3K36 and is upregulated by E7 [87]. Its overexpression is linked with cell proliferation and invasiveness and with poor prognosis in cervical cancer, candidating it as a biomarker and therapeutic target in this malignancy [87]. However, further studies are needed to investigate its role in head and neck cancers.

**- KDM2B** is a demethylase specific for the H3K4 and H3K36 residues and its deregulation has been associated to tumorigenesis [97]. It has been demonstrated that hr-HPV E6 and to a less extent E7 induce its overexpression in both human primary keratinocytes and cervical cancer cell lines through the downregulation of miR-146a-5p [88]. Thus, KDM2B seems to represent an important epigenetic player in HPV+ tumors [88].

All four members of the **KDM5 (A-D)** demethylase family catalyze the demethylation of the H3K4 histone mark and are functionally redundant; however, their activity on the modulation of the epigenetic profiles and of gene expression seems to be context-dependent [98].

**- KDM5A** is responsible for demethylating tri- and di-methyl groups from the H3K4 histone mark. Even in this case the HPV oncoprotein E7 exerts a crucial role in the modulation of its expression levels, and, in particular, it induces its upregulation. KDM5A overexpression represses the expression of miR-424–5 that downregulates the expression of SUZ12, a component of the PRC2 complex, further underlying the complex and intricate interplay among epigenetic regulators. KDM5A overexpression is oncogenic and correlated with aggressiveness and poor prognosis in cervical cancer [89].

**- KDM5B** is overexpressed in cervical cancers and in HNSCC compared to normal tissues [99,100]. High levels of KDM5B are associated with poor prognosis. Interestingly, KDM5B inversely correlates with STING expression levels and is associated with a suppressed immune response, low levels of tumor infiltrating lymphocytes, suggesting that, as observed for other epigenetic factors, targeting KDM5B could represent a novel promising strategy also by inducing an antitumor immune response [101].

**- KDM5C**, also known as JARID1C, is a histone H3K4 demethylase. It is a tumor suppressor and an X-linked gene, it is often mutated in tumors and in HPV+ HNSCC patients, and is less expressed in males compared to women in HNSCC [31,102]. E6 physically interacts with KDM5C leading to its E6AP- and proteasome-dependent degradation [90]. Upon interacting with E2, KDM5C is recruited to the long control region promoter (LCR) of the oncoviral genes E6 and E7, inducing their expression [91]. KDM5C inactivation is associated with failed heterochromatin assembly and genome instability, a mechanism that could be exploited by the virus for malignant transformation [31]. Moreover, HPV- mediated downregulation of KDM5C activates the expression of oncogenes such as EGFR and c-MET through the activation of cancer super-enhancers [31,90]. Moreover, KDM5C participates in STING silencing, thus suppressing antitumoral immune response [101], similarly to KDM5B.

**- KDM6A (UTX)** as well as **KDM6B (JMJD3)** are histone demethylases responsible for the demethylation of H3K27me3 and upregulated by E6 and E7 [55]. However, another study did not show the same effect of the oncoviral proteins on the regulation of KDM6B [56]. Interestingly, knockdown of KDM6A or KDM6B in HPV+ cervical cancer cell lines induces cell death and E7 expression induces a KDM6B dependence for cell growth [55,92]. KDM6A is described as a tumor suppressor in several cancers, but as an oncogene in others. It lies on the X-chromosome and it is associated with sex-differences in bladder cancer [103]. Overall, the exact role of this gene in HPV+ HNSCC has not been clearly elucidated, as well as its potential different role exerted in males and females HNSCC patients.

Finally, chromatin or histone regulators involved in the regulation of other hPTMs are emerging as promising targets in HNSCC pre-clinical research, for example BAP1 and RNF20/40. BAP1 catalyzes H2A deubiquitination inducing radioresistance: targeting BAP1 sensitizes both HPV− and HPV+ HNSCC to radiotherapy [104]. RNF20/40 is, instead, a ubiquitinase complex that mono-ubiquitinates histone H2B at lysine 120. H2Bub1 promotes an opened chromatin conformation accessible to transcription and DNA repair factors. RNF20/40 acts as a tumor suppressor and its loss-of-function and decreased activity have been associated with tumor progression, invasiveness and epithelial to mesenchymal transition (EMT). Interestingly the HPV viral protein L2 interacts with RNF20/40, and this interaction phenocopies the loss-of-function effects, promoting tumor growth and EMT [105]. This is an example of how HPV can affect the expression or activity of histone regulators responsible of also other hPTMs.

Altogether, these observations highlight how the E6 and E7 hr-HPV oncoviral proteins participate in the epigenetic reprogramming of the host cells, activating multiple oncogenic pathways. This suggests that targeting epigenetic regulators could represent a valuable and promising option for HPV+ HNSCC, also considering that, as discussed, several of these regulators are involved in the modulation of the viral proteins themselves. Thus, it is conceivable that HPV+ HNSCC, or some distinctive HPV+ subgroup [106,107], could be uniquely vulnerable to specific epi-drugs. Moreover, among all the oncogenic pathways, it is clear that E7 exerts a crucial role in dampening host innate immunity through different ways, including the regulation of epigenetic factors. Combining epi-drugs with immunotherapies is thus considered as another promising therapeutic strategy for HPV+ HNSCC.

## 5. Therapeutic Strategies in HPV+ and HPV− Head and Neck Squamous Cell Carcinoma: Present and Future Perspectives

### 5.1. Therapeutic Approaches for HPV+ and HPV− HNSCC

Despite the remarkable differences characterizing HPV+ and HPV− HNSCC, HNSCC patients are still treated with the same therapeutic strategies. HPV+ HNSCC are usually more responsive to chemo- and radiotherapies and are characterized by better prognosis. De-escalating strategies are thus being proposed with the aim to maintain efficacy while reducing acute and/or chronic toxicities. Recent evidence showed that treatment de-intensification leads to worse outcomes: current knowledges are insufficient to recommend changes in treatments based on HPV status [13,108]. However, it has been recently shown that, in selected patients, transoral robotic surgery (TORS) with de-escalating adjuvant radiotherapy is effective and less toxic, thus suggesting that further investigations could lay the basis to enroll specific patients in which this strategy could turn out as promising [109].

Overall, published studies suggest that the HPV status could be indicative for the prognosis, but to date it cannot represent a criteria towards the eligibility for specific treatments [110].

The main first-line therapeutic options in HNSCC are still, surgery, chemo- and radiotherapy, used both alone or in combination according to the specific clinical evaluation and stage of the disease [13,108,111]. Cetuximab, a FDA approved therapeutic, is an EGFR inhibitor which can be used in the treatment of HNSCC [13]. It acts by altering EGFR signaling and mediating antigen-specific immune response, thus leading to cell death. However, treatment with cetuximab is associated with only 13% response rate in relapsed and/or metastatic HNSCC patients [13]. On the other hand, combinatorial treatment of cetuximab with chemotherapy and radiotherapy has been shown to sensibly increase the response to chemo and radiotherapies alone [13]. Importantly in 2016, FDA approved the two anti-PD-1 (Programmed Cell Death 1) Immune Check-point Inhibitors (ICIs) Pembrolizumab and Nivolumab, for treating HNSCC patients [13]. Anti-PD-1 drugs show few side-effects and survival improvements, but even in this case durable survival benefits have been observed only in a relatively small group of patients [13]. HPV+ and HPV− HNSCC are among the cancers with the highest levels of immune infiltrates, however, cancer cells develop several mechanisms to elude immunosurveillance. Clinical trials suggest that in HNSCC patients the HPV status does not impact the response to immunotherapies [112].

Studies performed both in HNSCC as well as in other malignancies, suggest that combinatorial approaches utilizing immunotherapy agents and other drugs could represent a promising strategy [20,113]. Interestingly pre-clinical data are demonstrating that targeting epigenetic regulators, both broad and targeted reprogrammers such as DNMTs, HDACs, EZH2, LSD1 and KDM4A, leads to the activation of immune response pathways and boosts the anti-tumor immune response, suggesting that combining immune check-point inhibitors with specific epi-drugs could be a novel valuable promising therapeutic option to be studied and considered for HNSCC treatment [20,114,115]. To date in HNSCC significant disease benefits have been observed combining ICIs, as pembrolizumab, with chemotherapy and several preclinical studies and clinical trials are under evaluation to assess different combinatorial therapeutic strategies, also with epi-drugs [13,20,115,116,117,118,119]. Many other immune-based therapeutic approaches are in clinical trials as well as novel strategies aimed at targeting specifically the E6 and E7 oncoproteins [119]. Interestingly, strategies based on the cleavage of E6/E7 encoding genes from HPV DNA are tested in patients with cervical neoplasia, both through CRISPR/CAS9 and zinc-finger nuclease-based strategies [119].

Moreover, genetic and transcriptional profiling of the HNSCC subtypes is opening to the identification of novel targeted therapies. Several pre-clinical studies are ongoing, and some inhibitors are in clinical trials for specific cohorts of patients: for instance PI3K inhibitors are being tested, both alone or in combination with anti-PD-1 therapies in a panel of HPV-related cancers [119].

### 5.2. Epi-Drugs in HPV+ HNSCC

HPV− and HPV+ HNSCC, even though differently, carry several alterations in epigenetic pathways and profiles [119], implicating possible applications of epigenetic-based therapeutic strategies for these diseases.

Despite the important advances achieved in this field in the last decades, only few epigenetic therapeutics have been approved and are currently used in the clinics.

To date, there are nine FDA approved epi-drugs: four HDAC inhibitors (HDACi), two DNMT inhibitors (DNMTi), two isocitrate dehydrogenase (IDH) inhibitors and, recently, an EZH2 inhibitor, the first approved targeted epigenetic drug [30].

Although demethylating agents, such as the two DNMT inhibitors cytosine analogues 5-azacydine and 5-aza-2′-deoxycytidine (decitabine), and the second-generation hypomethylating prodrug SGI-110 (guadecitabine) gave different results in various solid tumors, for some liquid cancers such as acute myeloid leukemia and myelodysplastic syndromes, their clinical use has significantly improved patients’ quality of life and overall survival [30]. For the other malignancies several trials are ongoing [30].

Evidence is suggesting that DNA-methylation represents a promising target for the treatment of HNSCC, especially for the HPV+ subtype [120]. The DNMTi azacitidine and decitabine are in phase I or in phase II for HNSCC treatment, alone or in combination with other drugs, including immunotherapeutics [20,40]. It has been extensively demonstrated that DNMTi treatment in combination with chemotherapy is particularly effective, also in rescuing from cisplatin resistance [121]. Interestingly, HPV+ HNSCC cell lines seem more sensitive than the HPV− ones to DNMTi, and this could be in part due to the reduced expression of HPV genes, to the stabilization and reactivation of p53 and of active caspases in HPV+ HNSCC upon 5-azacytidine treatments [120]. Moreover, 5-azacytidine in particular, seems to activate type I IFN responses and to inhibit the invasive ability of HPV+ HNSCC cells [40].

The approved HDACi are Vorinostat (or suberoylanilide hydroxamic acid, SAHA), Belinostat (PXD-101), panobinostat (LBH589) and romidepsin (FK228, FR901228) [20]. The introduction of HDACi into the clinics has provided an important improvement in the field of epigenetic therapies. HDACi ameliorated the therapeutic management of some blood malignancies such as T-cell lymphomas and multiple myeloma, but seem less effective for solid tumors [30]. An important pitfall of these drugs is their low target specificity and their pleiotropic effects which lead to a vast array of side effects, and this has been also observed in HNSCC clinical trials [29,40]. Except for romidepsin, that is specific for class I HDACs, the other drugs are pan-inhibitors targeting, without specificity, the activity of multiple HDACs. Moreover, as for other epigenetic regulators, HDACs have also non-histone substrates, making it more difficult to predict all the molecular and systemic consequences of their inhibition [29]. Thus, more efforts are needed to further understand the mechanism of action of these enzymes in different contexts and to develop selective HDACi that could offer improved safety and efficacy.

Differently from DNMTi, HDACi have a lower efficacy when used as monotherapy [30]. Despite this, preclinical studies suggest that the use of HDACi in HNSCC is promising. These data are enforced by published and on-going clinical trials evaluating them both in monotherapies and in combination with EGFR inhibitors, chemo-, radio- and immunotherapies [40]. For example, the combination of HDACi (vorinostat or panobistat) with other drugs, such as erlotinib, cetuximab or cisplatinum, resulted to be clinically beneficial and tolerable [20,40]. However, a clinical trial in which HDACi were being evaluated in combination with chemotherapy was interrupted due the high toxicity, suggesting that more studies are needed to optimize the dosages for these therapies [40]. To note, a limit of these trials is that the HPV status and a differential response between HPV+ and HPV− HNSCC is rarely considered and evaluated, as reviewed in [40]. More efforts are thus required to design studies that take the HPV status into consideration.

The use of EZH2i in HNSCC is supported by several preclinical evidences. EZH2 is upregulated in HNSCC and is associated with immune evasion, metastasis and, more in general, with aggressiveness and poor prognosis [40]. Targeting EZH2 in HNSCC seems to represent a promising strategy and it should also sensitize to chemo and immunotherapies [40].

Clinical trials evaluating the use of EZH2 inhibitors in recurrent and metastatic HNSCC have started only in 2020 and data are not yet available.

In addition, pre-clinical data are indicating several other epigenetic enzymes as potential novel targets for HNSCC such as LSD1, BRD4, KDM6A, the NSD family members, alone or in combination with other approved epigenetic inhibitors, immunotherapies, chemo- or radiotherapies or targeted therapies, such as cetuximab or gefitinib [20,82,122,123,124].

To conclude, in HNSCC, epi-drugs pre-clinical data and clinical trials seem to be highly promising and several advances are expected in the next years. However, in general, more efforts are required in considering the HPV status in these studies.

## 6. Sex and Gender Bias in the Epigenetic Landscape of HPV+ HNSCC

Sex bias in tumor incidence, mortality and response to therapies is well documented across a large number of cancers, with males generally showing higher incidence and mortality, worse response to some therapeutic regimens and shorter post-treatment survivals compared to females [125,126]. Moreover, females are well known to have a stronger innate and adaptative immune response than males, thus contributing to reduced tumorigenesis and cancer progression [127]. Sex-bias is also emerging in immunotherapeutic treatments response [127]. Interestingly the epigenetic landscape and chromatin state has been demonstrated to be differentially influenced according to sex [128] and this could have important implications in diseases susceptibility and in epigenetic based therapies. Thus, taking into account the effect of sex in cancer research is of utmost importance [126].

Sex differences in cancer are the result of the lifelong interplay that occurs between sex, a biological factor related to the presence of XX and XY chromosomes in each cell of the body, and gender, intended as the sum of behavior, lifestyle, gender perception and gender norms of the individual [125]. Discerning between these two aspects is not easy and the reasons explaining sex-related differences in cancer are far from being clearly elucidated.

Overall, despite the recognized relevance of sex and gender in clinics, these variables are rarely included in preclinical research studies aimed at identifying novel markers and therapeutic targets or at investigating therapeutic response, cancer progression and survival.

However, in the last years the awareness of the importance of considering sex-bias is considerably increasing: an evident signal of this is the introduction, in 2016 by the National Institute of Health (NIH), of the duty to include sex among the biological variables in research studies [129]. The European Commission also has made it mandatory to discuss the sex and gender dimension on its proposal applications.

Significant sex-related differences have been also observed in HNSCC, where the male to female incidence ratio is globally approximately 3:1. Even though generally higher in males, this ratio can be different according to the geographical area. Importantly, this sex unbalance in HNSCC seems to be independent from the HPV status: indeed, the male to female ratio in HPV+ HNSCC spans from 3 to 6 [6] and, considering only OPSCC, of which approximately 60–70% are hr-HPV-positive tumors [7], the prevalence remains higher in men than in women. Moreover, women with HPV+ HNSCC seem to have improved overall survival compared to men [130].

The reasons underlying sex and gender differences in HPV+ HNSCC have not been fully elucidated. It could be either due to differences in sexual behavior between males and females, or that HPV transmission to the head and neck district is more effective from women to men than the opposite, but the topic is still debated [4].

Besides a possible difference in sexual behavior and gender lifestyle, a crucial role exerted by sex chromosome and hormones is also emerging in HNSCC. An interesting study published in 2016 and based on TCGA data, showed significative sex-bias differences in HNSCC at the mutational, mRNA and protein expression levels and in the DNA methylation patterns [131]. However, in this as well as in other studies, neither the HPV status nor the specific anatomical regions have been taken in consideration. Indeed, difficulties in studying the impact of sex and gender in HNSCC, beyond the limits linked to the lack of awareness and to set research practices, are further increased by an intrinsic high tumor heterogeneity. Sex-bias should be considered together with many variables as the HPV status, the alcohol and tobacco consumption and the anatomical regions affected. Indeed, as previously mentioned, these aspects are associated with clinical and molecular different profiles and can influence each other. Published studies generally take into account only one or few variables, and this is due also to statistical issues: indeed, considering all these criteria leads to very small groups of HNSCC patients, especially for women cohorts. As an example, in the previously mentioned study [131], upon subdividing the TCGA HNSCC cohort of 279 patients for both sex and HPV status, would result in a group of HPV+ HNSCC consisting of only 4 females [16,132].

Sex, as a biological factor, is the result of sex chromosomes and hormones.

Sex hormones and hormone receptors play a key role in several malignances, among which HNSCC. They are involved in tumorigenesis, in the modulation of tumor microenvironment, metabolism and immune response. Moreover, hormone receptors expression levels are used as biomarkers and therapeutic targets. Altered expression levels of hormone receptors such as the Androgen Receptor (AR) or the Estrogen Receptors, ERα and ERβ, have been found in HNSCC, but due to the high heterogeneity of these tumors, the hormone receptor expression pattern and its clinical significance in HNSCC is unclear. In HNSCC, evidences suggest that estrogens and progesterones are protective and exert a favorable role in females [133], that AR is associated with poor prognosis [134,135] and that there is a crosstalk between ERα and EGFR receptors that, when expressed together, are associated with bad prognosis and chemoresistance [136,137]. Interestingly, ERα expression seems to promote HPV integration and to be associated with a good prognosis in HPV+ OPSCC patients [138]. Therefore, sex-bias differences could be influenced by hormones in HPV+ HNSCC and should be taken in consideration.

Importantly, hormones also cross-talk with the cellular epigenetic machinery: they can influence sex differences in DNA methylation levels and chromatin accessibility in specific regions and with tissue specificity [125,128,139]. An intricate interplay between HPV and hormonal signaling pathways could thus conceivably differentially affect the epigenetic profiles in a sex-biased manner in HPV+ HNSCC.

Although sex hormones are considered one of the most important determinants of sexual dimorphisms, several studies have also shown the crucial role exerted by the different asset of chromosomes in male and female cells on the regulation of somatic gene expression [140]. Genetic and epigenetic sex differences are strictly correlated with the presence of two X chromosomes in females and a single copy of X and Y chromosomes in males. The Y-chromosome mainly harbors genes involved in male sex determination, in cell cycle regulation, signal transduction, protein stability and in the regulation of gene expression [141]. Loss of Y Chromosome (LoY) has been observed in approximately 25% of HNSCC: it occurs in both HPV− and HPV+ HNSCC but more frequently in HPV− HNSCC, maybe due to an association with smoking habits [142]. In HPV− HNSCC, LoY is associated with a worse overall survival and resistance to chemo- and radiotherapies [142]. Females have two copies of the X-chromosome. X-chromosome genes include sex determining genes, tumor suppressors, epigenetic regulators and genes involved in the immune response. To reduce the transcriptional dosage imbalance, in female cells the additional X chromosome, during early embryonic development, undergoes X-chromosome inactivation (XCI), an epigenetic driven phenomenon [102,143]. However, XCI is incomplete and a subset of X-chromosome genes, (in humans 15% of the X-linked genes) escape X inactivation. This phenomenon seems to protect females from complete loss of function due to single mutations and could explain, at least in part, the lower incidence of cancer in females. A comprehensive study analyzing TCGA data of 21 different tumors found that 6 of 783 X-linked genes are more commonly mutated in males than in females and are tumor suppressors. Interestingly these are also genes that escape X-inactivation (EXITS genes) and include *ATRX*, *CNKSR2*, *DDX3X*, *KDM5C*, *KDM6A* and *MAGEC3* [102]. Many of these are epigenetic regulators and, importantly, in some cases, for example *DDX3X* and *KDM6A*, the homologous genes located on the Y chromosome do not exert the same function of the X-linked ones [102].

Mutations or deregulation in the expression levels of X-linked genes encoding for epigenetic regulators could significantly affect the epigenetic profiles on autosomal chromosomes. Alterations in autosomal genes encoding for epigenetics factors could affect the proper X-chromosome inactivation which is indeed regulated by epigenetic mechanisms as DNA methylation and histone modifications that together with lncRNAs, drive chromosome condensation [143].

Overall, this intricate epigenetic crosstalk between autosomal and sex-chromosomes, can possibly lead to an epigenetic-mediated sex-biased cancer susceptibility.

Since HPV impacts on the epigenetic machinery and profiles of the host cell, it is conceivable that it could indirectly induce epigenetic instability in inactivated X chromosome (Xi), leading to molecular and clinicopathological sex-related difference in HPV+ HNSCC. Future studies are important to investigate these aspects. A few interesting observations in support of this hypothesis have been described. HPV indeed, influences the expression of some of the EXITS genes, for example of *DDX3X*, an ATP-dependent RNA-helicase whose downregulation mediated by E6 in HPV+ lung cancer is associated with poor prognosis [144,145], and of *KDM5C* and *KDM6A*, as previously described.

To conclude, it is desirable that future research aimed at designing novel epigenetic based therapeutic strategies for HNSCC would take into account not only the HPV status, but also the sex of the patients. In light of the differences between males and females in immune response pathways, it is also possible that the immune stimulation promoted by targeting epigenetic factors could be different based on sex, requiring specific efforts.

## Figures and Tables

**Figure 1 ijms-23-03483-f001:**
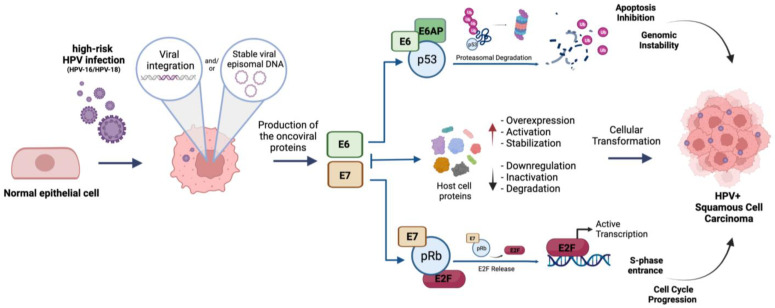
Schematic representation of hr-HPV (HPV16/18) mediated cellular transformation of infected epithelial cells. E6: oncoviral protein E6; E7: oncoviral protein E7; E6AP: ubiquitin ligase E6 associated protein; p53: tumor suppressor protein p53; pRb: oncosuppressor protein pRb; E2F: E2F-1 transcription factor. Created with BioRender.com.

**Table 1 ijms-23-03483-t001:** Table summarizing the main clinicopathological and molecular differences distinguishing the HPV− and the HPV+ HNSCC [1,4,6,7].

HPV−	Clinicopathological Aspects	HPV+
- Alcohol and tobacco	**Main risk factors**	- Sexual behavior
- Mainly oral cavity and larynx	**Anatomical subsite**	- Mainly oropharynx
- Higher	**Age of diagnosis**	- Lower (within the 6th decade of life)
- Modestly to well differentiated. - More keratinized	**Cellular differentiation** **status**	- Poorly differentiated. - Less keratinized
- Lower	**Immune infiltration**	- Higher
- Worse	**Response to conventional therapies**	- Better
- Less favorable: ~55%	**Prognosis: 5-year survival rates**	- More favorable: 75–80%
**HPV−**	**Molecular Features**	**HPV+**
- Frequently mutated	**p53**	- Generally WT - Degraded by the E6 hr-HPV oncoviral protein
- Low expressed	**p16Ink4a**	- Highly expressed
- High	**Mutational burden**	- Low
- Hypermutational status and chromosomal instability induced mainly by alcohol and tobacco carcinogens	**Factors mediating cellular transformation**	- E6 and E7 hr-HPV oncoproteins - Genomic rearrangements induced by viral genome integration
**HPV− and HPV+ HNSCC are characterized by** **different transcriptional and mutational profiles**		

**Table 2 ijms-23-03483-t002:** Modulation of hPTMs in HNSCC compared to normal tissues and in HPV+ HNSCC or E6/E7 transduced cells compared respectively to HPV− HNSCC or to “Cellular Control” (empty-vector transduced cells). ↑: upregulation; ↓: downregulation; –: control considered for the described sample comparison. ^1^ and ^2^ are used to indicate the samples considered when two options are reported in the column title. The technical approach (“Method”) through which the hPTMs have been detected is also reported.

hPTM	Normal Tissues ^1^ or Cellular Control ^2^	HPV+ HNSCC Tissues ^1^ or E6-E7 Transduced Cells ^2^	HPV− HNSCC Tissues	HNSCC Tissues (No Mention of HPV Status)	Method	Refs.
**H3K27me3**	– ^2^	↓ ^2^			Western Blot	[55]
**//**		↑ ^1^	–		IHC	[54]
**//**	– ^1^			↓	Mass Spectrometry	[53]
**H3K327me3/K36me1**	– ^1^			↓	Mass Spectrometry	[53]
**H3K327me2/K36me1**	– ^1^			↓	Mass Spectrometry	[53]
**H3K327me1/K36me1**	– ^1^			↑	Mass Spectrometry	[53]
**H3K327me1/K36me2**	– ^1^			↑	Mass Spectrometry	[53]
**H3K36me2**	– ^1^			↑	Mass Spectrometry	[53]
**//**	– ^1^			↑	IHC	[60]
**//**		– ^1^	No differences		IHC	[60]
**H3K36me1**	– ^1^			↑	Mass Spectrometry	[53]
**H3K14ac**	– ^1^			↓	Mass Spectrometry	[53]
**H4K20me1**		↑ ^1^	–		IHC	[54]
**H4K20me3**		↓ ^1^	–		IHC	[54]

**Table 3 ijms-23-03483-t003:** Hr-HPV E6/E7-mediated regulation of some of the main histone modifiers responsible for the deposition or removal of acetylated residues.

Histone Modifier	Function	Histone Target	hr-HPV Mediated Deregulation	Refs.
**p300/CBP**	Acetyltransferase	All four histone cores, p53, pRb and other non-histone targets	Both E6 and E7 interact with p300/CBP affecting its activity	[63,64,65]
**PCAF/KAT2B**	Acetyltransferase	H3 and H4 histones and non-histone targets	E7 interaction inhibits PCAF/KAT2B acetyltransferase activity	[66]
**TIP60/KAT5**	Acetyltransferase	H4 and H2A and non-histone targets	Destabilized by E6	[67,68]
**GCN5/KAT2A**	Acetyltransferase	H3, H4 and non-histone targets	Upregulated by E7	[69,70]
**HDAC1–** **HDAC2**	Deacetylase	Pan-Ac	E7 indirectly interact with HDAC1 and HDAC2, modulating their activity	[28,71,72]
**Sirt1**	Deacetylase	H1, H2A, H3, H4 and non-histone targets as p53 and FOXO proteins	Upregulated by E7	[73,74,75]

**Table 4 ijms-23-03483-t004:** Hr-HPV E6/E7-mediated regulation of histone modifiers responsible for the deposition or removal of methylated residues.

Histone Modifier	Function	Histone Target	hr-HPV Mediated Deregulation	Refs.
**EZH2**	Methyltransferase	H3K27me3	Upregulated by E6 and E7	[80,81,82,83]
**SUV39H1**	Methyltransferase	H3K9me3	Upregulated by E6 and E7	[84]
**NSD1/2/3**	Methyltransferase	H3K36me1/me2	HPV+ tumors have higher levels of NSD proteins than HPV−	[85]
**SET7**	Methyltransferase	H3K4, H3K37me1/2/3, H4K20me1and non-histone targets as p53	E6 interacts with SET7 and inhibits its activity	[86]
**KDM2A**	Demethylase	H3K36me1/2	Upregulated by E7	[87]
**KDM2B**	Demethylase	H3K36me2, H3K4me3	Upregulated by E6 and E7	[88]
**KDM5A**	Demethylase	H3K4me2/me3	Upregulated by E7	[89]
**KDM5C**	Demethylase	H3K4me2/me3	Proteasomal degradation induced by E6	[31,90,91]
**KDM6A (UTX)**	Demethylase	H3K27me3	Upregulated by E7	[55,56]
**KDM6B (JMJD3)**	Demethylase	H3K27me3	Maybe upregulated by E7 (controversial results)	[55,56,92]

## Data Availability

Not applicable.

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
