# Peer review of "High Risk-Human Papillomavirus in HNSCC: Present and Future Challenges for Epigenetic Therapies"

_ijms, 2022, doi:10.3390/ijms23073483_

Round 1

Reviewer 1 Report

The primary goal of the review is to provide an updated overview on the impact of Human Papillomavirus (HPV) infections on the histone post-translational modifications (hPTMs) landscape in Head and Neck Squamous Cell Carcinoma (HNSCC). The review furnishes a good and generalized background that summarizes the current state of the topic. In addition, the review describes in detail, epigenetic therapeutic strategies and approaches for HNSCC treatment, highlighting the more efforts required to identify new biomarkers for detecting HPV+ and HPV- HNSCC for whose it might be possible used epi-drugs. The topic is clinically relevant since infection with human papillomavirus (HPV) is associated with a variety of cancer. The review is well written and fluent in reading. The manuscript is not just a descriptive summary of the topic but presents critical discussions with contradictory researches in the area of focus, including elements of debate and presenting both sides of the argument with a good grade-appropriate word patterns and vocabulary. Finally, there are  some minor and major concerns that need to be addressed.

Minor concerns:

  • Lane 70-73: the sentence is too long. It needs to be restructured.
  • Lane 93: Please add dot at the end.
  • Lane 95: the sentence is not clear: it should be read ranging from 88-93%
  • Lane 99: I believe higher should be replace with lower, otherwise the sentence makes no sense.
  • Lane 122: modification should be read modifications
  • Lane 127: please add dot at the end.
  • Lane 132: since later the authors used hr as abbreviation for high risk, here high risk should be read high risk (hr)
  • Lane 133: Abbreviations should be spelt just the first time they are mentioned. In this line head and neck carcinoma should be read HNSCC.
  • Lane 166: among which should be read including
  • Lane 192 e 200: histone…. Should be read hPTM
  • Lane 238-240: the sentence is redundant
  • Lane 263: therapeutics targtes identification should be read therapeutic target identification
  • Lane 284: Histone deacetylase (HDAC) should be read just HDAC
  • Lane 286: HDACs should be read HDAC
  • Lane 318: landscapes in cancers. [43-45]. Should be read landscapes in cancers [43-45].
  • Lane 330 and 341: “Immunohistochemistry” should be read IHC
  • Lane 336: all hPTMs should be reported in Upper letter.
  • Lane 367-369: The sentence needs to be revised.
  • Lane 372: ASEs should be read ASE
  • Lane 385: should be reported the paragraph “4.1.1” instead of “4.1.2”.
  • Lane 386-396-401: histone….. should be read HAT
  • Lane 403-404: the sentences are not clear.
  • Lane 407: Interestingly, TIP60 acetylates histone H4 that is then recognized 406 by BRD4, which subsequently represses HPV E6 transcriptional activation. I would suggest to restructure the sentence: Interestingly, TIP60 acetylates histone H4 that is recognized by BRD4. The latter represses HPV E6 transcriptional activation.
  • In Table 3 Legend it should mention that are acetylated residues at the end of the sentence.
  • Lane 422 HDACs (Histone Deacetylases) should be read HDACs
  • Lane 423: the sentence is not clear
  • Lane 426: HIF-1 should be read Hypoxia inducible factor (HIF-1). Later lane 427 HIF-1a (Hypoxia inducible factor 1a)  should be read HIF-1a.
  • Lane 430: IFNb should be read IFNβ
  • Legend of Table 4 should be revised.
  • Lane 463: the authors should specify that it is “DNMT3A” related to reported reference.
  • Lane 564-566: the sentence is incomplete. I would add and stage of the disease.
  • Lane 566: an FDA should be read a FDA
  • Lane 567:  Cetuximab , an FDA approved therapeutic, is an EGFR inhibitor which can 566 be used in the treatment of EGFR-overexpressing HNSCC. The sentence is incorrect. Use of cetuximab in HNSCC does not require EGFR overexpression.
  • Lane 569 with only a 13% should be read with only 13%
  • Lane 587: it is not clear whether these durgs refer to ICI or epigenetic drug. Whether they refer to ICI, to the best of my knowledge no clinical evidence are available. In any case please add appropriate references.
  • Lane 663, 690: adjust correct position of comma in the sentences.

Major concerns:

  • Among histone post-translational modifications (hPTMs) reported in the text, I suggest to take in consideration histone ubiquitination and its relationship with tumor arise. Indeed, Eckhardt et al. demonstrated that the L2 HPV protein physically interacts with the RNF20/40 histone ubiquitination complex and promotes tumor cell invasion in an RNF20/40-dependent manner (see reference: Eckhardt et al., “Multiple Routes to Oncogenesis Are Promoted by the Human Papillomavirus-Host Protein Network,” Cancer Discov., vol. 8, no. 11, pp. 1474–1489, Nov. 2018)
  • Lane 576: The paper “[106]”, the authors refer to, does not report that HPV+ HNSCC are characterized by higher levels of immune infiltrates indicating a less immunosuppressive tumor microenvironment in comparison to the HPV- ones, on the contrary, both HPV-positive and -negative HNSCCs belong to cancer types with the highest immune infiltration.

Author Response

The primary goal of the review is to provide an updated overview on the impact of Human Papillomavirus (HPV) infections on the histone post-translational modifications (hPTMs) landscape in Head and Neck Squamous Cell Carcinoma (HNSCC). The review furnishes a good and generalized background that summarizes the current state of the topic. In addition, the review describes in detail, epigenetic therapeutic strategies and approaches for HNSCC treatment, highlighting the more efforts required to identify new biomarkers for detecting HPV+ and HPV- HNSCC for whose it might be possible used epi-drugs. The topic is clinically relevant since infection with human papillomavirus (HPV) is associated with a variety of cancer. The review is well written and fluent in reading. The manuscript is not just a descriptive summary of the topic but presents critical discussions with contradictory researches in the area of focus, including elements of debate and presenting both sides of the argument with a good grade-appropriate word patterns and vocabulary. Finally, there are  some minor and major concerns that need to be addressed.

Our Answer: Thank you very much for your comments.

Minor concerns:

  • Lane 70-73: the sentence is too long. It needs to be restructured.
  • Lane 93: Please add dot at the end.
  • Lane 95: the sentence is not clear: it should be read ranging from 88-93%
  • Lane 99: I believe higher should be replace with lower, otherwise the sentence makes no sense.
  • Lane 122: modification should be read modifications
  • Lane 127: please add dot at the end.
  • Lane 132: since later the authors used hr as abbreviation for high risk, here high risk should be read high risk (hr)
  • Lane 133: Abbreviations should be spelt just the first time they are mentioned. In this line head and neck carcinoma should be read HNSCC.
  • Lane 166: among which should be read including
  • Lane 192 e 200: histone…. Should be read hPTM
  • Lane 238-240: the sentence is redundant
  • Lane 263: therapeutics targtes identification should be read therapeutic target identification
  • Lane 284: Histone deacetylase (HDAC) should be read just HDAC
  • Lane 286: HDACs should be read HDAC
  • Lane 318: landscapes in cancers. [43-45]. Should be read landscapes in cancers [43-45].
  • Lane 330 and 341: “Immunohistochemistry” should be read IHC
  • Lane 336: all hPTMs should be reported in Upper letter.
  • Lane 367-369: The sentence needs to be revised.
  • Lane 372: ASEs should be read ASE
  • Lane 385: should be reported the paragraph “4.1.1” instead of “4.1.2”.
  • Lane 386-396-401: histone….. should be read HAT
  • Lane 403-404: the sentences are not clear.
  • Lane 407: Interestingly, TIP60 acetylates histone H4 that is then recognized 406 by BRD4, which subsequently represses HPV E6 transcriptional activation. I would suggest to restructure the sentence: Interestingly, TIP60 acetylates histone H4 that is recognized by BRD4. The latter represses HPV E6 transcriptional activation.
  • In Table 3 Legend it should mention that are acetylated residues at the end of the sentence.
  • Lane 422 HDACs (Histone Deacetylases) should be read HDACs
  • Lane 423: the sentence is not clear
  • Lane 426: HIF-1 should be read Hypoxia inducible factor (HIF-1). Later lane 427 HIF-1a (Hypoxia inducible factor 1a)  should be read HIF-1a.
  • Lane 430: IFNb should be read IFNβ
  • Legend of Table 4 should be revised.
  • Lane 463: the authors should specify that it is “DNMT3A” related to reported reference.
  • Lane 564-566: the sentence is incomplete. I would add and stage of the disease.
  • Lane 566: an FDA should be read a FDA
  • Lane 567:  Cetuximab , an FDA approved therapeutic, is an EGFR inhibitor which can 566 be used in the treatment of EGFR-overexpressing HNSCC. The sentence is incorrect. Use of cetuximab in HNSCC does not require EGFR overexpression.
  • Lane 569 with only a 13% should be read with only 13%
  • Lane 587: it is not clear whether these durgs refer to ICI or epigenetic drug. Whether they refer to ICI, to the best of my knowledge no clinical evidence are available. In any case please add appropriate references.
  • Lane 663, 690: adjust correct position of comma in the sentences.

 Our Answer: We addressed all your concerns: you can assess all modifications in track changes. With regards to your comment

  • Lane 99: I believe higher should be replace with lower, otherwise the sentence makes no sense

We deleted the sentence.

Major concerns:

  • Among histone post-translational modifications (hPTMs) reported in the text, I suggest to take in consideration histone ubiquitination and its relationship with tumor arise. Indeed, Eckhardt et al. demonstrated that the L2 HPV protein physically interacts with the RNF20/40 histone ubiquitination complex and promotes tumor cell invasion in an RNF20/40-dependent manner (see reference: Eckhardt et al., “Multiple Routes to Oncogenesis Are Promoted by the Human Papillomavirus-Host Protein Network,” Cancer Discov., vol. 8, no. 11, pp. 1474–1489, Nov. 2018)
  • Lane 576: The paper “[106]”, the authors refer to, does not report that HPV+ HNSCC are characterized by higher levels of immune infiltrates indicating a less immunosuppressive tumor microenvironment in comparison to the HPV- ones, on the contrary, both HPV-positive and -negative HNSCCs belong to cancer types with the highest immune infiltration.

Our Answer: thank you for these insightful comments we added what requested (paragraph starting at lane1131) and changed lane 576- current lane 1221.

Reviewer 2 Report

The article was interesting and well-organized. There were a few minor typos and I would suggest some minor edits to the language as several sections proved somewhat difficult to read. I have highlighted some of the suggested changes below:

Line 44 - "of" should replace "in"

Table 1A - I believe this should be 6th rather than 6 degrees as written

Line 60 - capitalization inconsistent with remainder of table

Line 67 - "for" should replace "of"

75 - consider revising

81/82 - consider revising

91 - 95 - should this be 2 sentences?

145 - 146 - not sure if this should be cervices or if you are referring to different areas of the same cervix

153 - 161 - consider revision of paragraph

229 - evidence is emerging?

234 - 236 - consider revision

254 - should this be pattern

257 - "with" rather than "to"

263 - consider revision of sentence

270 - unsure if this was its own paragraph

285 - has rather than have

295 - with rather than to

368 - 369 - "which" or "that"

424 - revise for clarity (is necessary?)

469 - revise for clarity 

470   relatively should be relative

543 - did you mean to say through?

586, 595, 716-717 - consider revising for clarity

There were several places in the manuscript where the spacing appeared a bit off. I am not sure if this was due to the way it rendered on the website but you may want to check the manuscript for spacing errors.

Author Response

The article was interesting and well-organized.

Our answer: thank you very much for your postive comment.

There were a few minor typos and I would suggest some minor edits to the language as several sections proved somewhat difficult to read. I have highlighted some of the suggested changes below:

Line 44 - "of" should replace "in"

Table 1A - I believe this should be 6th rather than 6 degrees as written

Line 60 - capitalization inconsistent with remainder of table

Line 67 - "for" should replace "of"

75 - consider revising

81/82 - consider revising

91 - 95 - should this be 2 sentences?

145 - 146 - not sure if this should be cervices or if you are referring to different areas of the same cervix

153 - 161 - consider revision of paragraph

229 - evidence is emerging?

234 - 236 - consider revision

254 - should this be pattern

257 - "with" rather than "to"

263 - consider revision of sentence

270 - unsure if this was its own paragraph

285 - has rather than have

295 - with rather than to

368 - 369 - "which" or "that"

424 - revise for clarity (is necessary?)

469 - revise for clarity

470   relatively should be relative

543 - did you mean to say through?

586, 595, 716-717 - consider revising for clarity

Our Answer: we believed we have addressed all typos.

The only exception is lane 285 ( current lane 550) we left have because the sentence now reads : "Altered expression and/or activity of HATs and HDACs have been described in a wide range of malignancies [29]. "

There were several places in the manuscript where the spacing appeared a bit off. I am not sure if this was due to the way it rendered on the website but you may want to check the manuscript for spacing errors.

Our answer: we checked spacing errors, thank you.

Reviewer 3 Report

The manuscript “High Risk-Human Papillomavirus in HNSCC: present and future challenges for epigenetic therapies” is a review article regarding the impact of HPV on the histone post-translational modifications. The manuscript is generally well written, easy to read and conclusions are clearly stated.

However, authors should address some points before the manuscript could be considered for publication.

Major

Figure 1: in the legend please specify the name of proteins involved; for example “E6: oncoviral protein E6”.

In the paragraph “epigenetics in cancer” authors did not mention the biomarker Nicotinamide N-methyltransferase (NNMT) that is upregulated in Head and Neck Squamous Cell Carcinoma (PMID: 29882109). NNMT-expressing cancer cells have an altered epigenetic state that includes hypomethylated histones and other cancer-related proteins (PMID: 23455543) and there is an increasing evidence demonstrating the involvement of this enzyme in chemoresistance (PMID: 34018676, PMID: 32150326).

Line 279, 291 and everywhere else in the text: sometimes bold is used only for the first part of the sentence, sometimes for the whole sentence. This kind of formatting is confusing. I suggest to use a short bold title as first sentence.

This paper would benefit from showing results of analysis of mRNA expression of some elements (e.g. HMTs and HDMs..) in HNSCC using publicly available datasets e.g., RNA seq data.

Minor

Please carefully read and correct typos.

Line 127: a dot is missing at the end of the sentence.

Table 1: “transcriptionaland” should be spaced.

Author Response

The manuscript “High Risk-Human Papillomavirus in HNSCC: present and future challenges for epigenetic therapies” is a review article regarding the impact of HPV on the histone post-translational modifications. The manuscript is generally well written, easy to read and conclusions are clearly stated.

Our Answer: thank you for  postive comment.

However, authors should address some points before the manuscript could be considered for publication.

Major

Figure 1: in the legend please specify the name of proteins involved; for example “E6: oncoviral protein E6”.

Our Answer: thank you we did.

In the paragraph “epigenetics in cancer” authors did not mention the biomarker Nicotinamide N-methyltransferase (NNMT) that is upregulated in Head and Neck Squamous Cell Carcinoma (PMID: 29882109). NNMT-expressing cancer cells have an altered epigenetic state that includes hypomethylated histones and other cancer-related proteins (PMID: 23455543) and there is an increasing evidence demonstrating the involvement of this enzyme in chemoresistance (PMID: 34018676, PMID: 32150326).

Our answer: Thank you for suggesting this important information.  We added  a new paragragraph starting at line 346.

Line 279, 291 and everywhere else in the text: sometimes bold is used only for the first part of the sentence, sometimes for the whole sentence. This kind of formatting is confusing. I suggest to use a short bold title as first sentence.

Our answer: we  adrressed this issue, thank you.

This paper would benefit from showing results of analysis of mRNA expression of some elements (e.g. HMTs and HDMs..) in HNSCC using publicly available datasets e.g., RNA seq data.

Our Answer: thank you we added this information starting at line 793.

Minor

Please carefully read and correct typos.

Line 127: a dot is missing at the end of the sentence.

Table 1: “transcriptionaland” should be spaced.

Our Answer: thank you we did.